# ShareGPT4Video: Improving Video Understanding and Generation with Better Captions

**Lin Chen**[1,4*] **Xilin Wei**[4*] **Jinsong Li**[2,4*] **Xiaoyi Dong**[2,4*] **Pan Zhang**[4] **Yuhang Zang**[4]
**Zehui Chen**[1,4] **Haodong Duan**[4] **Bin Lin**[3] **Zhenyu Tang**[3]
**Li Yuan**[3] **Yu Qiao**[4] **Dahua Lin**[2,4,5] **Feng Zhao**[1†] **Jiaqi Wang**[4†]
[1] USTC [2] CUHK [3] PKU [4] Shanghai AI Lab [5] CPII under InnoHK

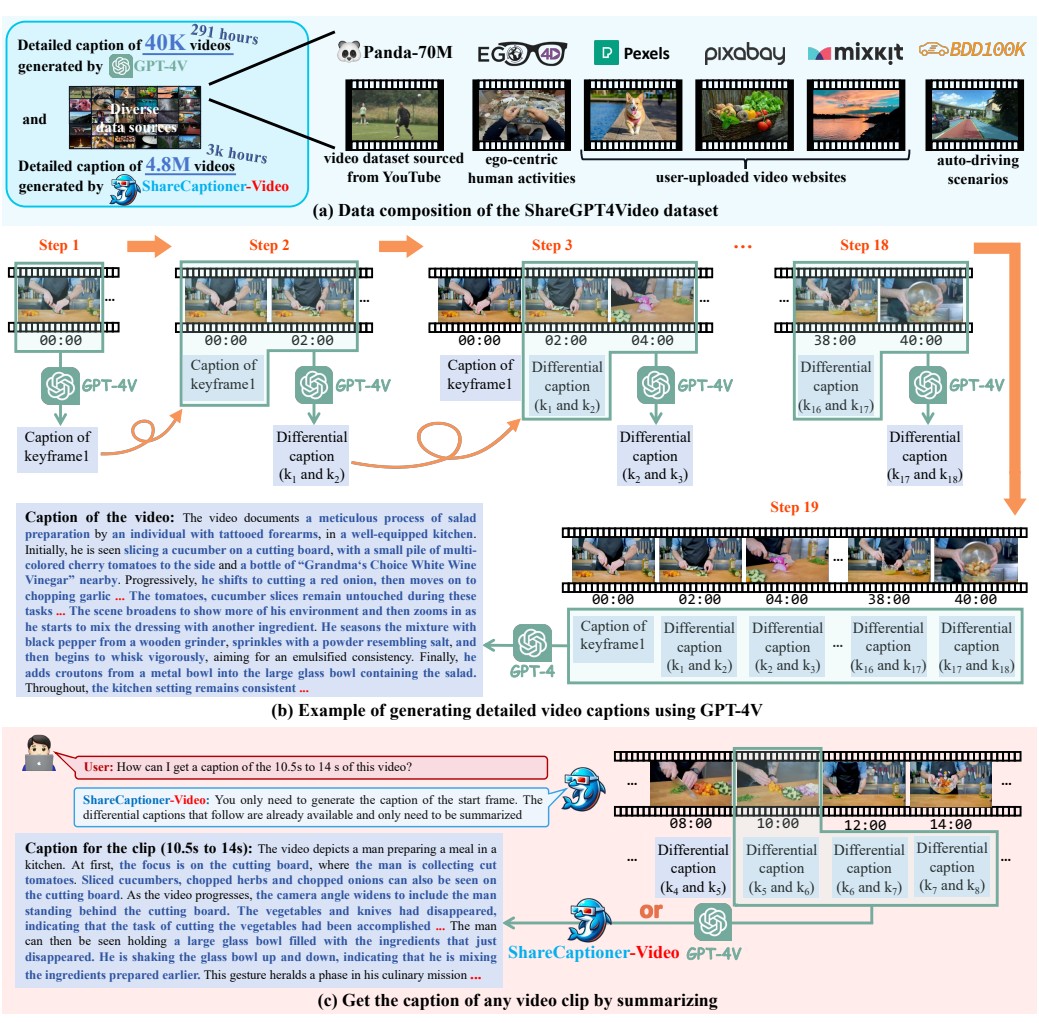

Figure 1: **Details and attributes of the ShareGPT4Video.** (a) The proposed ShareGPT4Video dataset contains a large volume of high-quality video-caption pairs collected from diverse sources, with 40K captions from GPT4V and 4.8M captions from our ShareCaptioner-Video. (b) We illustrate in detail the process of harnessing the multi-modal image model GPT4V [54] to generate high-quality captions for videos. Please refer to Figure 10 for the full caption of the example. (c) Our unique captioning strategy enables the re-caption of sub-clips by reusing their differential captions.

*Equal Contribution. † Corresponding author.

38th Conference on Neural Information Processing Systems (NeurIPS 2024) Track on Datasets and Benchmarks.

## Abstract

We present the ShareGPT4Video series, aiming to facilitate the video understanding of large video-language models (LVLMs) and the video generation of text-to-video models (T2VMs) via dense and precise captions. To achieve this, taking aside the non-scalable costly human annotators, we find using GPT4V to caption video with a naive multi-frame or frame-concatenation input strategy leads to less detailed and sometimes temporal-confused results. We argue the challenge of designing a high-quality video captioning strategy lies in three aspects: **1) Inter-frame precise temporal change understanding. 2) Intra-frame detailed content description. 3) Frame-number scalability for arbitrary-length videos.** To this end, we meticulously designed a differential video captioning strategy, which is stable, scalable, and efficient for generating captions for videos with arbitrary resolution, aspect ratios, and length. Based on it, we construct ShareGPT4Video, which contains 40K high-quality videos spanning a wide range of categories, and the resulting captions encompass rich world knowledge, object attributes, camera movements, and crucially, detailed and precise temporal descriptions of events. Based on ShareGPT4Video, we further develop ShareCaptioner-Video, a superior captioner capable of efficiently generating high-quality captions for arbitrary videos. We annotated 4.8M aesthetically appealing videos by it and verified their effectiveness on a 10-second text2video generation task. For video understanding, we verified the effectiveness of ShareGPT4Video on several current LVLM architectures and presented our superb new LVLM ShareGPT4Video-8B. All the models, strategies, and annotations will be open-sourced and we hope it can serve as a pivotal resource for advancing both the LVLMs and T2VMs community. We released the full project at `https://sharegpt4video.github.io/`.

## 1 Introduction

Recent advancements in multi-modal learning, driven by large language models, have led to progress in image-text dialogue[41, 11, 18, 66, 76, 15, 2, 78, 90, 89] and text-to-image generation tasks[3, 61, 7, 27, 63, 62, 84, 9]. This has inspired a shift towards video understanding[40, 1, 38, 83, 48, 50, 75, 47] and generation tasks[64, 23, 70, 67, 5, 45], allowing for user interaction across video and language modalities. Thus, the detailed and high-fidelity video captions, which bridge the aforementioned modalities, are instrumental in propelling the advancements within the field.

Despite the rich semantic and temporal content of videos, they are often paired with brief captions in existing data. These short descriptions limit the detailed video understanding and the controllability of video generation. While the importance of detailed captions is recognized in image-text dialogue[11, 8, 65] and text-to-image generation tasks[9, 4], similar efforts are lacking in video understanding and generation.

However, creating large-scale, high-quality video captions is challenging. Detailed captioning for long videos is non-trivial and time-consuming even for humans, hindering large-scale annotation. Current open-source LVLMs lack this capability, and closed-source APIs do not yet support video inputs. On the other hand, if we roughly degrade the input from video to multiple frames, even GPT4V struggles to describe the video with satisfied quality. For example, an intuitive idea is to provide multiple frames with timestamps to the GPT4V and generate the caption, while we find that GPT4V is unstable and sometimes misunderstands the temporal relation between the frames, and its performance further degrades with the increasing of video frames. Others such as concatenating all the frames into a large image are non-helpful to the temporal problem, and the caption loses details as the frame number increases. We also showcase these problems in Figure 11 and 12.

We posit that the challenge of devising an effective video captioning strategy is rooted in three fundamental aspects: *1) Inter-frame precise temporal change understanding*: The temporal dimension distinguishes videos from images. An imprecise temporal description can significantly diminish the quality of the video caption and lead to confusion in the training models. *2) Intra-frame detailed content description*: Detailed descriptions [11] are crucial for aligning modalities between image and text, which are also important for video-text alignment. *3) Frame-number scalability for arbitrary-*

*length videos*: Videos encountered in the wild can vary greatly in length. An ideal captioning strategy should be resilient to this variability and generate appropriate captions for videos of any length.

To this end, we present the **Differential Sliding-Window Captioning strategy** (DiffSW), which is *stable, scalable, and efficient for generating captions for arbitrary videos.* The central concept of DiffSW is translating the all-frames-to-caption task into a differential description task. Specifically, we generate a detailed caption for the first frame and apply a sliding window of length two to the subsequent frames in chronological order. The powerful image multi-modal model, GPT4V [54], is tasked with identifying the changes between frames based on three inputs: the previous frame, its differential caption, and the current frame. This encompasses alterations in camera movement, object movement, character actions, and scene transitions. Upon acquiring all differential captions, these are input into GPT4 [53] to construct a comprehensive caption for the entire video. The differential concept allows DiffSW to concentrate on the changes between frames, i.e., the temporal changes. Its sliding design ensures the correctness of temporal order and invariance towards the total number of frames. The constant input frame number guarantees that GPT4V does not overlook details and utilizes the API efficiently, resulting in stable, scalable, and efficient caption quality from DiffSW. Furthermore, the differential design enables the re-caption of any sub-clips of a captioned video by reusing its differential captions.

Based on DiffSW, we construct **ShareGPT4Video**, which contains **40K high-quality video-caption pairs** spanning a wide range of categories, and the resulting captions encompass rich world knowledge, object attributes, camera movements, and crucially, detailed and precise temporal descriptions of events. The videos of ShareGPT4Video are collected from various sources [14, 79, 56, 21, 57, 51], employed with a Semantic-based Data Filtering strategy to mitigate content homogeneity among these videos. A Semantic-aware Key-frame Extraction strategy is then applied to the videos to reduce the temporal redundancy. DiffSW is applied to the keyframes to generate high-quality captions and we further improve its stability and quality with a Hierarchical Prompt Design. Manual quality inspection is employed to ensure the quality of the video captions.

Based on ShareGPT4Video, we present ShareCaptionor-Video, an exceptional video captioner capable of efficiently generating high-quality captions for videos of a wide range of resolution, aspect ratio, and duration. It enables the further scaling of high-quality video caption data with minor cost and satisfactory quality, and we generate high-quality captions for 4.8M aesthetically appealing videos (totaling about 3000 hours) by it.

We conduct extensive experiments in video understanding and generation tasks to demonstrate the value of our high-quality video-caption dataset and our superior video captioner. For video generation, a DiT-based [55] text-to-video model trained on the 4.8M video-captions pairs performs well in generating 10-second high-resolution videos and achieving fine-grained control over content generation. For video understanding, ShareGPT4Video brings consistent performance gain of multiple current LVLMs over multiple benchmarks by replacing a small proportion of training data. We further present ShareGPT4Video-8B, a simple yet superb LVLM that reached SOTA performance on three advancing and comprehensive video benchmarks. The model, strategy, and annotations will be publicly available and we hope this project can serve as a pivotal resource for advancing both the LVLMs and T2VMs community.

## 2 Related Work

**Large video-language models.** The vision-language community has recently made significant advancements in the field of large image-language models [42, 41, 43, 11, 15, 12, 68, 69, 2, 82, 71, 20, 25, 26, 73, 46, 28, 92, 85, 58, 87, 39]. These models typically employ three core components: a visual encoder for extracting visual features, a connector to bridge the visual and language modalities, and a large language model (LLM) to decode the multimodal context. A milestone in this field is LLaVA-1.5 [41], which uses CLIP-Large [59] as the visual encoder, a two-layer MLP as the connector, and Vicuna-v1.5 [16] as the LLM backbone. This model achieves exceptional performance with only affordable data for training. Following this paradigm, the large video-language model field has also seen emerging efforts to encode video into vision tokens suitable for LLMs, enabling understanding and reasoning about video content. For example, VideoChat2 [36] employs a video transformer and a Q-Former [34] to extract and compress visual tokens. VideoLLaVA [40] uses a pre-aligned video encoder with image and language, along with an MLP, to extract and transform visual tokens.

LLaMA-VID [38] introduces the LLaVA-like three-stage training strategy and compresses each frame's vision token into one to support long video understanding. Despite the continuous exploration of training strategies and model architectures, none of these methods have investigated the role of high-quality captions in aligning video and language modalities, as has been done in the large image-language model field [11, 8].

**Text-to-video models.** Recent advancements in the text-to-image field, such as DALL·E [3], and Stable Diffusion [61], have significantly propelled the T2I area. Inspired by these achievements, researchers have increasingly begun to explore the potential of generating high-fidelity videos from textual descriptions. For instance, MagicVideo [88] introduces a 3D U-Net-based architecture and directs temporal attention to ensure efficient, high-fidelity video generation while maintaining temporal consistency and authenticity. PixelDance [81] and SVD [5] leverage pixel-based and latent-based techniques to produce high-resolution videos, while Make-A-Video [64] and Imagen Video [22] extend text-to-image models and techniques to the text-to-video domain. A milestone in this field is Sora [45], which ambitiously employs DiT to train a text-to-video model from scratch. Sora is the first model capable of generating minute-long videos based on user instructions. The success of this remarkable feat relies on extensive high-quality video-caption data, meticulously designed architectural optimizations, and substantial computational power. Despite the pivotal role of high-quality video-caption data in T2VMs, this aspect has not received sufficient attention. Therefore, in this work, we aim to construct a high-quality video-caption dataset to advance the development of T2VMs.

## 3  ShareGPT4Video Dataset

This section provides a detailed exposition of how we construct the ShareGPT4Video dataset. We detail the entire process in Figure 2.

### 3.1  Data Collection

**Selection of Data Sources.** To serve both video understanding and video generation tasks, we consider the aesthetic quality and content complexity of videos during our collection process. We first consider Panda-70M [14], a high-resolution video dataset sourced from YouTube, featuring clips ranging in one minute. This open-domain source covers diverse areas such as wildlife, cooking, sports, news & TV shows, gaming & 3D rendering. It typically includes complex content and transitions, providing a solid foundation for understanding various real-world scenarios. However, the complexity of these contents and transitions presents a significant challenge for the video generation field. To address this, we also source a large volume of aesthetically appealing videos from some user-uploaded video websites [56, 57, 51]. These videos predominantly consist of scenic views and aesthetically pleasing human activities, involving fewer transitions and simpler events. Finally, we supplement our collection with selected videos from Ego4D [21] and BDD100K [79] to fill the gaps in ego-centric human activities and auto-driving scenarios, ensuring our video sources encompass as many real-world scenes as possible.

**Semantic-Based Data Filtering.** Although our captioning method can support videos of extended lengths, our collection primarily focuses on videos shorter than two minutes due to the trade-off of the duration and amount of videos. We initially filter out videos from our selected data sources longer than two minutes, leaving videos in two minutes as the candidates. We then introduce a semantic-based data filtering strategy to mitigate content homogeneity among these candidates and maintain diversity in the final video dataset. This approach aims to select videos with significant thematic differences from the pool of candidates to compose our final video collection. Specifically, we first use the Panda-Student [14] model to generate a short caption with one sentence for each candidate video, and then maintain a final pool of video candidates. Whenever a new video $V$ is processed, we encode its corresponding short caption $S$ using the Bert-Base-Uncased [17] language model to obtain the CLS token $P_{n+1} \in \mathbb{R}^{1 \times D}$, which captures high-level semantic expressions. We then calculate the similarity between this CLS token $P_{n+1}$ and the CLS tokens $\{P_1, P_2, \ldots, P_n\}$ of videos already in the final candidate pool. A new video will only be added to the pool if its maximum similarity is below a predefined threshold. We provide the pseudo-code in Figure 14.

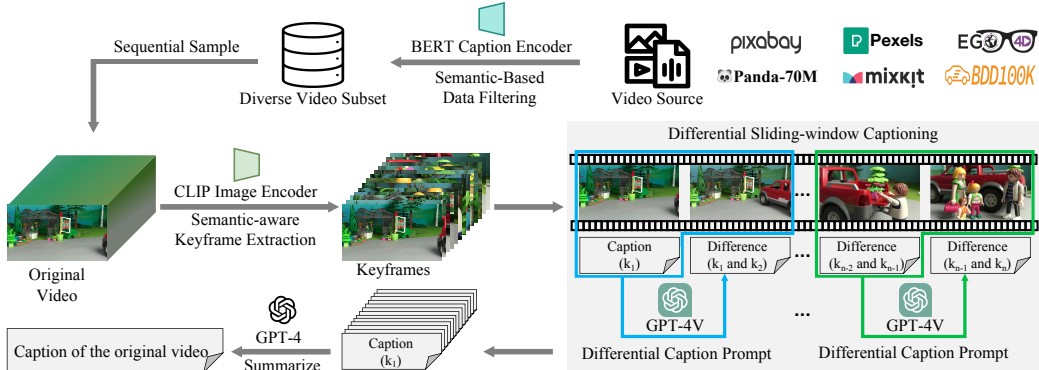

Figure 2: **Pipeline for generating high-quality video-caption data.** We begin by selecting diverse video sources based on aesthetic quality and content complexity. Next, we use semantic-based data filtering to prevent content homogenization. We then apply semantic-aware key-frame extraction for sparse sampling, maintaining significant semantic variations. Finally, we implement a differential sliding-window captioning strategy, utilizing GPT-4V to generate detailed and temporally rich captions.

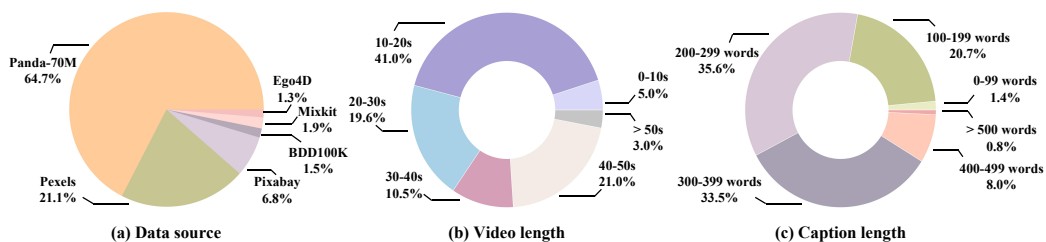

Figure 3: **Comprehensive video-caption dataset**: (a) The dataset covers a broad spectrum of content, including wildlife, cooking, sports, scenery, ego-centric human activities, auto-driving scenarios, etc. (b) The dataset includes videos ranging from 2 seconds to 2 minutes in length. (c) The captions primarily range from 200 to 400 words, providing rich temporal information that serves video understanding and generation tasks well.

## 3.2 Video Processing

Videos are commonly redundant on the temporal dimension, and keyframe sampling is a general idea to represent a video compactly. However, traditional key-frame extraction methods [91, 6] often struggle to ensure semantic coherence, leading to missing key-frames covering crucial changes and transitions. Consequently, we develop a semantic-aware key-frame extraction method that strikes a balance between reducing temporal redundancy in videos and maintaining the semantic coherence of the content.

**Semantic-aware Key-frame Extraction.** We denote $V \in \mathbb{R}^{T \times H \times W \times 3}$ as a $T$ frame set sampled from a video with fixed 2-second intervals. We calculate the keyframe set $V' \in \mathbb{R}^{T' \times H \times W \times 3}$ that are sufficiently sparse yet comprehensively cover the evolution of events within the video that $T' < T$. We view the output CLS token of the CLIP-Large image encoder [59] as the global semantics of each frame and remove the adjacent frames that have a high semantic similarity. In practice, we initialize the keyframe set $V'$ with the first frame of $V$. For each frame in $V$, we calculate its semantic similarity $d$ with the latest keyframe in $V'$. If $d$ is lower than the pre-defined threshold, we view the frame as a keyframe and add it to the $V'$. If not, the frame is skipped as redundant. For completeness, the last frame of $V$ is always added in $V'$. We provide the pseudo-code in Figure 15.

## 3.3 Captioning Pipeline

As we mentioned in Section 1, we find if we feed all the frames to the GPT4V directly, the GPT4V struggles to stably generate captions with the correct temporal relation between frames, and its performance further worsens with the frame number increasing. On the other hand, if we concatenate all the frames into a large image, the GPT4V loses more details with the increasing frame number, as shown in Figure 11 and 12. To this end, a **stable**, **scalable**, and **efficient** strategy is essential for large-scale annotation of videos with arbitrary length.

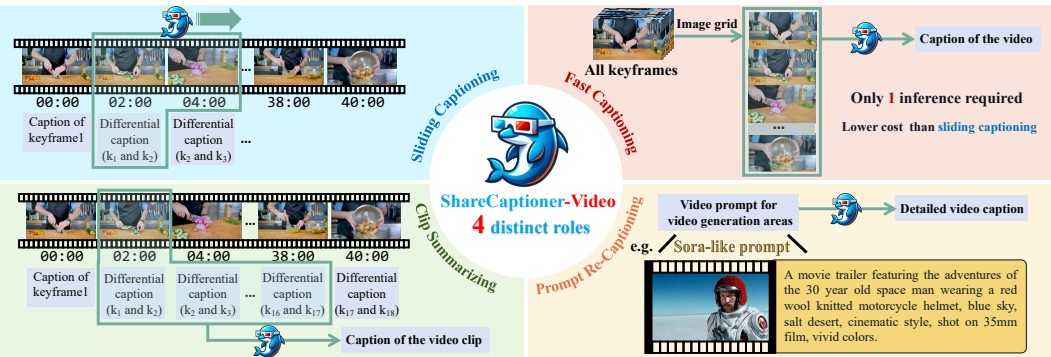

Figure 4: The ShareCaptioner-Video is a Four-in-One exceptional video captioning model with the following capabilities: Fast captioning, Sliding Captioning, Clip Summarizing, and Prompt Re-Captioning.

**Differential Sliding-window Captioning.** To this end, we develop a differential sliding-window captioning pipeline to generate high-quality captions with detailed temporal descriptions for various videos. Specifically, the input fed to the image multi-modal model each time includes the current key-frame and the previous key-frame along with its differential caption. Then, we introduce the Differential Prompt to guide GPT4V in focusing on the changes between the current and previous frames, such as posture, position, camera angle, etc. Additionally, incorporating the differential caption of the previous frame as supplementary context enhances the response quality and reduces hallucinations. This is because the image embedding and textual caption provide explicit and implicit representations of the image, respectively. The differential caption not only adds extra context but also integrates temporal information from two frames ago, further improving the model's temporal understanding. It's important to note that for the first key-frame, which lacks a preceding frame, its differential caption is replaced directly with the standard caption. Finally, we input all differential captions along with their corresponding timestamps into GPT4. A specific Summary Prompt is designed to instruct the LLM to generate high-quality video captions with precise temporal dynamics and detailed spatial information. In practice, we use `GPT-4-Turbo-04-09` for all the annotations.

In the design of the prompts, we discovered that an explicit Hierarchical Prompt Design significantly aids the GPT4V in comprehending its role, its expected format, and its operational boundaries. This approach contributes to the stabilization of the output's format and enhances the overall quality of the results. For more details, please refer to Section A.2

## 4 ShareCaptioner-Video

### 4.1 Model Design

We fine-tune the IXC2-4KHD [19] using the collected video caption data, resulting in our ShareCaptioner-Video. For flexible usage, we re-organize the data for the following capabilities:

**1. Fast Captioning** The model employs an image-grid format for direct video captioning, providing rapid generation speeds that are ideal for short videos. In practice, we concatenate all the keyframes of a video into a vertically elongated image and train the model on a caption task.

**2. Sliding Captioning** The model supports streaming captioning in a differential sliding-window format, yielding high-quality captions that are suitable for long videos. Similar to the captioning pipeline used in Section 3.3, we take the two adjacent keyframes alongside the previous differential caption as input, and train the model to describe the events occurring between them.

**3. Clip Summarizing** The model can swiftly summarize any clip from ShareGPT4Video or videos that have undergone the differential sliding-window captioning process, eliminating the need to re-process frames. We use all the differential descriptions as input, and the output is the video caption.

**4. Prompt Re-Captioning:** The model can rephrase prompts input by users who prefer specific video generation areas, ensuring that T2VMs trained on high-quality video-caption data maintain format alignment during inference with their training. In practice, we use GPT-4 to generate Sora-style prompts for our dense captions, and we train the re-captioning task in reverse, *i.e.*, by using the generated prompt as input and the dense caption as the training target.

Table 1: **The gain from high-quality captions is universal among model architectures and scales.** We report the baseline based on their public checkpoints. The best results are **bold**.

| Model | VideoBench | MVBench | TempCompass | Avg. |
|---|---|---|---|---|
| VideoLLaVA-7B [40] | 34.5 | 43.0 | 50.6 | 42.7 |
| VideoLLaVA-7B+Ours | **35.2** | **43.6** | **52.7** | **43.8** |
| LLaMA-VID-7B [38] | 36.5 | 41.3 | 48.1 | 42.0 |
| LLaMA-VID-7B+Ours | **38.2** | **43.2** | **50.6** | **44.0** |
| LLaMA-VID-13B [38] | 48.3 | 43.3 | 51.4 | 47.7 |
| LLaMA-VID-13B+Ours | **52.4** | **44.2** | **53.3** | **50.0** |

Table 2: **Combined with VQA data, detailed captions can benefit LVLMs more compared to short captions.** The baseline (first row) utilizes only 153K VQA data. The best results are in **bold**.

| Caption | Unlock ViT | VideoBench | MVBench | TempCompass | Avg. |
|---|---|---|---|---|---|
| – | ✗ | 37.3 | 47.2 | 57.2 | 47.2 |
| short | ✗ | 36.9 | 47.5 | 56.1 | 46.8 |
| short | ✓ | 37.5 | 47.9 | 56.9 | 47.4 |
| detailed | ✗ | 40.7 | 50.3 | 60.7 | 50.6 |
| detailed | ✓ | **41.2** | **51.2** | **61.5** | **51.3** |

In practice, we fine-tune the model end-to-end over one epoch. We follow the default high-resolution strategy, using 'HD-55' for fast captioning and 'HD-25' for the others. The learning rate is uniform across all model components and warms up from 0 to $2.5 \times 10^{-5}$ within the first 1% of steps. The batch size is set to $1024$, and we sample the data uniformly.

## 4.2 Scaling-up Captions

To validate the effectiveness of our ShareCaptioner-Video in the video captioning task and further support the development of the video generation domain, we utilized it to annotate a large volume of aesthetically appealing videos. Specifically, we meticulously collect and process 4.8 million video clips, totaling approximately 3000 hours, from three sources: MixKit [51], Pexels [56], and Pixabay [57]. Subsequently, we employ the sliding captioning mode of ShareCaptioner-Video to generate high-quality captions for these videos. The total captioning process requires approximately 4000 H100 GPU hours. We provide some statistics on generated captions in Figure 8.

## 5 Experiments

### 5.1 Video Understanding

**Datasets and Benchmarks.** To thoroughly explore the benefits that our high-quality video-caption data bring to LVLMs, we conduct comprehensive evaluations of the model across three multi-modal video benchmarks. VideoBench [52] curates approximately 15,000 QA pairs spanning 10 evaluation dimensions from 13 existing data sources, such as MSVD-QA [74], MSRVTT-QA [74], Activitynet-QA [80], etc. MVBench [36] is designed to challenge LVLMs with video tasks that cannot be effectively resolved by single-frame reliance, featuring 4,000 QA pairs derived from 11 public video benchmarks. TempCompass [44] specifically assesses the nuanced performance of LVLMs across various temporal aspects, such as speed, direction, and attribute changes. It includes 410 videos and 7,540 meticulously collected instructions, emphasizing temporal comprehension and interaction.

**Improving current LVLMs with ShareGPT4Video.** We validate the effectiveness of the high-quality video-caption data collected in ShareGPT4Video to improve the performance of current LVLMs. For fairness and simplicity, we integrate 28K high-quality video-caption data related to complex scenes (Panda-70M [14], Ego4D [21], and BDD100K [79]) of ShareGPT4Video to replace the captions data in the VideoChatGPT-100K [50] conversation data with an equivalent number. Then we train the VideoLLaVA [40] and LLaMA-VID [38] with their default training settings and hyperparameters.

As shown in Table 1, ShareGPT4Video consistently improves the alignment between video and language modalities in different LVLM architectures and scales. Specifically, VideoLLaVA-7B [40] achieves an average performance gain of 1.1 across three comprehensive multi-modal video benchmarks after integrating high-quality captions, while LLaMA-VID-7B and LLaMA-VID-13B achieve an average gain of 2.0 and 2.3, separately. Our high-quality video-caption data is particularly effective in helping LVLMs achieve significant performance improvements on benchmarks that require complex temporal understanding, such as TempCompass [44].

**ShareGPT4Video-8B.** To obtain our final ShareGPT4Video-8B model, we start with the LLaVA-Next-8B [32] image multi-modal model, implemented by the Open-LLaVA-Next codebase [13]. Consistent with previous LVLM approaches [40, 50], we uniformly sample 16 frames from each video and arrange these frames into a 4x4 image grid to form the input for both training and inference, following the IG-VLM [30] strategy. For training data, we first collect 153K VQA data from various instructional video-to-text datasets to build our baseline. This collection includes 13K conversational

Table 3: **Comparison with SOTA methods on TempCompass.** With 7B parameters, ShareGPT4Video-8B outperforms competitors in 19 out of 20 dimensions, despite these competitors using larger training data or more parameters. The best results are **bold** and the second-best results are underlined.

| Model | Multi-Choice QA | | | | | Yes/No QA | | | | | Caption Matching | | | | | Caption Generation | | | | | Avg. |
|---|---|---|---|---|---|---|---|---|---|---|---|---|---|---|---|---|---|---|---|---|---|
| | AC | DI | SP | EV | AT | AC | DI | SP | EV | AT | AC | DI | SP | EV | AT | AC | DI | SP | EV | AT | |
| Valley-7B [47] | 47.0 | 29.3 | 32.5 | 18.9 | 29.9 | 58.1 | 52.0 | 52.5 | 50.3 | 52.9 | 65.0 | 53.8 | 52.6 | 53.0 | 53.8 | 54.0 | 31.0 | 32.7 | 34.2 | 41.4 | 33.4 |
| PandaGPT-13B [14] | 35.5 | 27.8 | 29.3 | 31.8 | 30.9 | 53.0 | 49.6 | 50.8 | 53.7 | 52.2 | 56.6 | 51.4 | 44.3 | 55.0 | 49.0 | 23.7 | 25.7 | 26.0 | 29.8 | 32.6 | 40.4 |
| VideoLLaMA-13B [83] | 54.1 | 24.5 | 28.1 | 32.8 | 28.5 | 68.1 | 46.0 | 48.8 | 51.8 | 50.9 | 73.1 | 47.4 | 47.1 | 52.0 | 48.3 | 54.3 | 21.3 | 13.9 | 38.5 | 33.9 | 43.3 |
| VideoChatGPT-7B [50] | 47.0 | 31.6 | 28.4 | 37.1 | 30.9 | 52.5 | 50.0 | 49.5 | 51.0 | 50.0 | 64.6 | 48.6 | 47.8 | 49.3 | 48.6 | 40.9 | 28.4 | 24.5 | 31.8 | 33.9 | 42.4 |
| mPLUG-Owl-7B [76] | 66.6 | 29.3 | 32.2 | 34.8 | 35.4 | 64.4 | 50.6 | 51.2 | 51.3 | 52.0 | 56.9 | 45.3 | 46.4 | 49.3 | 49.0 | 46.5 | 28.2 | 30.4 | 31.2 | 36.5 | 44.5 |
| VideoLLaVA-7B [36] | 70.4 | 32.2 | 38.2 | 41.4 | 39.9 | 74.3 | 51.8 | 50.3 | 49.2 | 51.1 | 88.2 | 53.8 | 61.9 | 57.0 | 58.3 | 50.8 | 28.7 | 23.2 | 38.2 | 33.6 | 49.9 |
| LLaMA-VID-7B [38] | 58.6 | 29.9 | 29.3 | 30.5 | 26.0 | 63.0 | 48.8 | 49.2 | 48.4 | 52.7 | 72.7 | 45.6 | 52.2 | 49.0 | 49.0 | 53.0 | 28.0 | 21.9 | 35.5 | 35.9 | 44.2 |
| ShareGPT4Video-8B | 87.6 | 34.6 | 47.5 | 62.9 | 64.2 | 75.2 | 53.8 | 58.6 | 66.5 | 65.6 | 93.3 | 58.1 | 58.8 | 75.0 | 75.3 | 79.8 | 32.6 | 36.6 | 50.8 | 53.4 | 61.5 |

Table 4: **Comparison with SOTA methods on VideoBench.** * denotes our evaluation results with the public checkpoints. The best results are **bold** and the second-best results are underlined.

| Model | ANet | MSVD | MSRVTT | TGIF | YC2 | UCF | MOT | TV | MV | NBA | LE | DM | SQA3D | Avg. |
|---|---|---|---|---|---|---|---|---|---|---|---|---|---|---|
| mPLUG-Owl-7B [76] | 41.5 | 42.5 | 36.3 | 31.7 | 27.1 | 22.8 | 27.8 | 24.0 | 30.2 | 25.1 | 33.3 | 51.0 | 32.0 | 33.2 |
| Otter-7B [33] | 44.3 | 55.0 | 47.0 | 34.3 | 32.7 | 22.4 | 16.7 | 27.7 | 37.1 | 34.3 | 52.8 | 48.7 | 29.7 | 37.5 |
| Video-LLaMA-7B [83] | 39.9 | 41.2 | 34.1 | 31.3 | 28.9 | 27.6 | 16.7 | 24.8 | 32.4 | 26.2 | 60.6 | 49.1 | 31.2 | 32.8 |
| Valley-7B [47] | 38.1 | 32.0 | 28.0 | 31.4 | 29.1 | 20.3 | 11.1 | 23.7 | 32.6 | 31.3 | 41.7 | 56.5 | 33.3 | 34.0 |
| VideoChat-7B [35] | 44.6 | 42.2 | 37.4 | 33.7 | 27.7 | 22.4 | 27.8 | 26.2 | 34.1 | 28.6 | 39.9 | 55.4 | 31.4 | 35.4 |
| PandaGPT-7B [14] | 45.0 | 50.4 | 44.6 | 29.7 | 33.0 | 33.0 | 16.7 | 27.9 | 37.1 | 31.1 | 41.7 | 56.0 | 30.8 | 37.5 |
| VideoChatGPT-7B [50] | 46.6 | 57.5 | 46.3 | 35.6 | 34.8 | 24.1 | 27.8 | 28.8 | 36.5 | 22.5 | 41.7 | 58.2 | 37.2 | 38.5 |
| ChatUniVi-7B [29] | 49.0 | 48.6 | 41.7 | 41.3 | 29.0 | 28.3 | 16.7 | 23.1 | 33.6 | 25.7 | 38.9 | 53.1 | 29.1 | 35.3 |
| VideoLLaVA-7B* [40] | 44.1 | 34.5 | 30.0 | 39.4 | 30.7 | 19.5 | 22.2 | 27.3 | 33.4 | 25.6 | 33.3 | 50.7 | 38.9 | 34.5 |
| LLaMA-VID-7B* [38] | 45.2 | 44.5 | 39.1 | 29.1 | 29.3 | 27.9 | 11.1 | 34.1 | 32.5 | 28.9 | 36.1 | 47.8 | 36.8 | 36.5 |
| ShareGPT4Video-8B | 50.8 | 45.6 | 43.0 | 42.8 | 34.6 | 39.7 | 22.2 | 31.9 | 34.0 | 30.5 | 41.7 | 53.6 | 42.9 | 41.2 |

data from VideoChatGPT [50] and 140K question-answer pairs, with 45K data points from CLEVRER [77], 8K from EGO-QA [21], 34K from NextQA [72], and 53K from TGIF-Transition [37]. Then, these VQA data are combined with 28K video-caption data, forming a consolidated training dataset of 181K samples. For more training details, please refer to Section A.1.

As illustrated in Table 3, 4, and 5, we present a quantitative comparison between our ShareGPT4Video-8B model supercharged by our ShareGPT4Video dataset with existing state-of-the-art LVLMs. Notably, compared with previous LVLMs, our ShareGPT4Video-8B attains the most superior performance in all three comprehensive benchmarks. Specifically, thanks to the rich temporal information provided by ShareGPT4Video, our ShareGPT4Video-8B model achieves an impressive average accuracy of 61.5% on the TempCompass benchmark. This is an 11.6% increase over the previous best-performing LVLM, VideoLLaVA-7B. Additionally, despite the VideoBench and MVBench benchmarks collecting a diverse range of QA data from various existing video datasets, we achieve solid performance on these benchmarks, surpassing the previous state-of-the-art by an average accuracy of 2.7% and 8.2%.

**Ablation on caption quality and ViT.** Based on ShareGPT4Video-8B, we study how the modality alignment is influenced by caption quality and learnable vision encoder. As indicated in Table 2, introducing short captions on top of VQA data may not yield substantial performance gains. It could even degrade performance on some benchmarks due to sub-optimal modality alignment. Comparing the first, second, and fourth rows of Table 2, the significant performance gains of comprehending temporal sequences benefited from our high-quality caption data are evident. Moreover, unlocking the vision encoder when training with detailed captions facilitates better LVLMs modality alignment.

## 5.2 Video Captioning

To verify the capability of ShareCapitoner-Video, we quantitatively compare the video captioning quality between ShareCapitoner-Video and GPT4V with human preference voting. As shown in Table 7, it performs on par with GPT4V. We also shows the qualitative results in Figure 9. For more details, please refer to Section A.4

## 5.3 Video Generation

**Model setup.** To validate the effectiveness of high-quality captions in the T2VMs area, we utilize ShareCaptioner-Video and Panda-Student [14] to generate high-quality and short video captions for 4.5M videos with 65 frames and 0.3M videos with 221 frames, separately. Following the process outlined in the Open-Sora-Plan [31], we fine-tuned the pretrained T2VM to enable the generation

Table 5: **Comparison with SOTA methods on MVBench.** * denotes our evaluation results with the public checkpoints. The best results are **bold** and the second-best results are underlined.

| Model | AS | AP | AA | FA | UA | OE | OI | OS | MD | AL | ST | AC | MC | MA | SC | FP | CO | EN | ER | CI | Avg. |
|---|---|---|---|---|---|---|---|---|---|---|---|---|---|---|---|---|---|---|---|---|---|
| Otter-7B [33] | 23.0 | 23.0 | 27.5 | 27.0 | 29.5 | 53.0 | 28.0 | 33.0 | 24.5 | 23.5 | 27.5 | 26.0 | _28.5_ | 18.0 | 38.5 | 22.0 | 22.0 | 23.5 | 19.0 | 19.5 | 26.8 |
| mPLUG-Owl-7B [76] | 22.0 | 28.0 | 34.0 | 29.0 | 29.0 | 40.5 | 27.0 | 31.5 | 27.0 | 23.0 | 29.0 | 31.5 | 27.0 | 40.0 | 44.0 | 24.0 | 31.0 | 26.0 | 20.5 | 29.5 | 29.7 |
| LLaMA-Adapter [86] | 23.0 | 28.0 | 51.0 | 30.0 | 33.0 | 53.5 | 32.5 | 33.5 | 25.5 | 21.5 | 30.5 | 29.0 | 22.5 | 41.5 | 39.5 | 25.0 | 31.5 | 22.5 | 28.0 | 32.0 | 31.7 |
| VideoChatGPT-7B [50] | 23.5 | 26.0 | _62.0_ | 22.5 | 26.5 | 54.0 | 28.0 | 40.0 | 23.0 | 20.0 | 31.0 | 30.5 | 25.5 | 39.5 | _48.5_ | 29.0 | 33.0 | 29.5 | 26.0 | 35.5 | 32.7 |
| VideoLLaMA-7B [83] | 27.5 | 25.5 | 51.0 | 29.0 | 39.0 | 48.0 | 40.5 | 38.0 | 22.5 | 22.5 | 43.0 | 34.0 | 22.5 | 32.5 | 45.5 | _32.5_ | 40.0 | _30.0_ | 21.0 | _37.0_ | 34.1 |
| VideoChat-7B [35] | 33.5 | 26.5 | 56.0 | 33.5 | 40.5 | 53.0 | 40.5 | 30.0 | 25.5 | 27.0 | 48.5 | 35.0 | 20.5 | 42.5 | 46.0 | 26.5 | 41.0 | 23.5 | 23.5 | 36.0 | 35.5 |
| VideoLLaVA-7B* [40] | _46.0_ | **42.5** | 56.5 | 39.0 | 53.5 | 53.0 | _48.0_ | **41.0** | _29.0_ | _31.5_ | 82.5 | **45.0** | 26.0 | _53.0_ | 41.5 | **33.5** | _41.5_ | 27.5 | 38.5 | 31.5 | _43.0_ |
| LLaMA-VID-7B* [38] | 45.5 | _40.5_ | 58.0 | _39.5_ | 55.0 | 53.5 | 40.0 | 35.5 | 18.5 | 27.5 | **87.0** | _41.5_ | 23.0 | 45.5 | 41.0 | 27.0 | 40.0 | **34.5** | **41.5** | 31.5 | 41.3 |
| ShareGPT4Video-8B | **49.5** | 39.5 | **79.5** | **40.0** | _54.5_ | **82.5** | **54.5** | 32.5 | **50.5** | **41.5** | _84.5_ | 35.5 | **62.5** | **75.0** | **51.0** | 25.5 | **46.5** | 28.5 | _39.0_ | **51.5** | **51.2** |

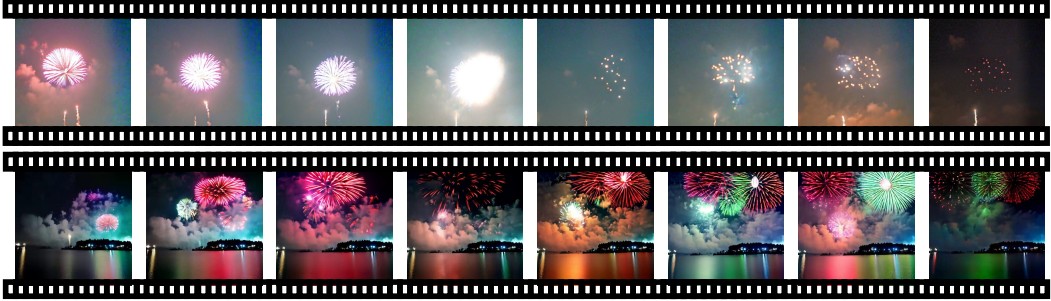

Prompt: A drone camera circles around a beautiful historic church built on a rocky outcropping along the Amalfi Coast, the view showcases historic and magnificent architectural details and tiered pathways and patios, waves are seen crashing against the rocks below as the view overlooks the horizon of the coastal waters and hilly landscapes of the Amalfi Coast Italy, several distant people are seen walking and enjoying vistas on patios of the dramatic ocean views, the warm glow of the afternoon sun creates a magical and romantic feeling to the scene, the view is stunning captured with beautiful photography.

Figure 5: **Example of 10-second text-to-video task.** The T2VM trained on the detailed video-caption data can exhibit impressive camera control.

Prompt: The video captures the spectacle of a continuous fireworks show against the backdrop of a starry night sky. It commences with a burst of vibrant reds, greens, purples, and yellows that paint the heavens and cast shimmering reflections upon the water below. As the display progresses, the fireworks evolve, transitioning from the initial array to a focus on radiant oranges, yellows, and fiery reds. These explosions form captivating clusters at the heart of the sky, ascending in breathtaking formations accompanied by trailing plumes of smoke, adding a dramatic flourish to the visual narrative. Throughout the duration, the fireworks maintain their dynamic allure, their patterns and positions evolving to underscore the ongoing spectacle. Meanwhile, the mirrored reflections on the water's surface faithfully echo the colors and shapes above, further enhancing the mesmerizing and ever-changing nature of the display.

Figure 6: **Influence of T2VM training caption length.** Thanks to the high-quality captions generated by ShareCaptioner-Video, the T2VM trained on the detailed video-caption data exhibits impressive semantic content control (video below), while the T2VM with short captions failed to follow the complex prompt (video above).

of high-fidelity 10-second videos. For comparison, we fine-tuned a baseline model with the same quantity of video-short-captions pairs. For more training details, please refer to Section A.1.

**Qualitative analysis.** As illustrated in Figure 5, the T2VM can accurately follow detailed prompts and demonstrate remarkable control over semantic content and camera movement when aided by high-quality, detailed captions generated by ShareCaptioner-Video. The resulting video showcases intricate and lively content. In contrast, when provided with brief captions, the T2VM struggles to adhere to complex generation prompts, leading to subpar results.

## 6 Limitations and Social Impacts

**Limitations.** Although our current pipeline for generating high-quality video captions fully utilizes visual and textual information, it is limited by GPT4V's inability to incorporate audio information

simultaneously. Audio information is beneficial in conversational scenarios involving daily human activities. We plan to introduce audio information in future work, once GPT4o supports audio input, to enhance the quality of our captions further. Additionally, the sampling interval for initial sparsification of the original videos and the window length setting in DiffSW was empirically set to 2 seconds and adjacent 2 frames based on the majority of videos. We plan to make these hyperparameters adaptive to video content in future work to handle a wider variety of video content effectively.

**Social impacts.** 1) Since the large language model involves the generation process of the large-scale captions, we have not manually verified each caption for socially biased content; 2) Although we utilize video data from existing public datasets, we cannot ensure that the selected videos do not contain human faces. Therefore, while there are no restrictions on the use of our generated captions, users must adhere to the licenses of the original video sources when using the videos. Our models can be manipulated or "jailbroken" to produce outputs that are non-inclusive or disrespectful as many LVLMs do. This vulnerability highlights the importance of continuing to improve the robustness and ethical alignment of LVLMs to prevent misuse and ensure they contribute positively to diverse applications.

# 7    Conclusion

In this study, we aim to address the challenge of lacking high-quality video-caption data for large video-language models (LVLMs) and text-to-video models (T2VMs). We develop ShareGPT4Video, a high-quality video-caption dataset, and ShareCaptioner-Video, an advanced and versatile model in the video-language multi-modal area. By employing a series of strategies and designs, we generate 40K detailed captions from advanced image multi-modal model, GPT4V, and 4.8M high-quality captions from our ShareCaptioner-Video. These captions include rich world knowledge, object attributes, camera movements, and detailed temporal descriptions of events. Our extensive experiments validate the effectiveness of our dataset and captioner in enhancing video understanding and generation tasks. We believe that ShareGPT4Video and ShareCaptioner-Video will serve as essential resources for advancing research in the LVLM and T2VM communities.

# 8    Acknowledgments

This work was supported by the Anhui Provincial Natural Science Foundation under Grant 2108085UD12. We acknowledge the partial support of the GPU cluster built by MCC Lab of Information Science and Technology Institution, USTC. This work was also partially supported by the Shanghai Artificial Intelligence Laboratory, the National Key R&D Program of China (2022ZD0160201), the Centre for Perceptual and Interactive Intelligence (CPII) Ltd under the Innovation and Technology Commission (ITC)'s InnoHK. Dahua Lin is a PI of CPII under the InnoHK.

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

# A  Appendix

In the appendix, we provide more results and analysis and summarize them as follows:

- In Section A.1, we detail the experimental setups.
- In Section A.2, we talk about the prompt design philosophy and showcase the prompt for each stage.
- In Section A.3, we present the template of our hierarchical prompt.
- In Section A.4, we introduce the qualitative and quantitative comparison of the caption quality between our ShareCaptioner-Video and other methods.
- In Section A.5, we present more statics of the ShareGPT4Video dataset.
- In Section A.6, we provide the detailed pseudo code of semantic-based data filtering and semantic-aware key-frame extraction.

## A.1  Experimental Details

**Training details of ShareGPT4Video-8B.** During training, we employ a batch size of 128 and the AdamW optimizer. We opt to fine-tune the entire model, setting the learning rate for the vision encoder at 2e-6, for the MLP projector at 2e-5, and for the LLM using LoRA [24], the learning rate is set at 2e-4. Such training strategy enables us to obtain an exceptional LVLM, ShareGPT4Video-8B, with 8 A100 GPUs in about 5 hours.

**Implementation details for T2VMs.** We utilize the Latte-XL [49] model with pre-trained weights and a text encoder from T5-XXL [60]. In the first stage, we perform pretraining on a lower number of frames and enable joint image-video training, with the image batch size being four times that of the video. This stage involved 50k training steps. In the second stage, both the video and image batch sizes were reduced to 2. For all training stages, we used AdamW optimizer with a constant learning rate of 2e-5, and the resolution for both images and videos was set to 512×512. For training precision, we used Bf16 since fp16 led to loss becoming NaN. The first stage requires around 6K H100 GPU hours, and the second stage requires around 36K Ascend GPU hours.

## A.2  Hierarchical Prompt Design.

We introduce this design to help the multi-modal and language models effectively perform their roles during the captioning process. We separately illustrate the differential-caption prompt and summary prompt in Figure 7 and Figure 8. Hierarchical prompts primarily consist of four components. The Character part provides the model with an overall perception of the role it is to play and the work environment it faces. The Skills section specifies the skills the model needs to possess, ensuring precise compliance with multiple requirements without omissions or confusion. The Constraints section clarifies behaviors that users do not desire and the rules to be followed when constructing outputs. The Structured Input section requires users to set up according to their specific scenarios. For example, when guiding the model to generate differential captions, the Character part informs the model that it is an expert in analyzing video frames. The Skills section requires the model to describe inter-frame target actions and behaviors, changes in environment and background, alterations in target appearance attributes, and camera movements reflecting temporal changes. The Constraints section demands precise descriptions without listing them item by item. In the Structured Input section, the input consists of frame indexes, timestamps, the previous frame, and its differential caption, among others.

## A.3  Prompt Template

**The Differential Caption Prompt Template**

# Character
You are an excellent video frame analyst. Utilizing your incredible attention to detail, you provide clear, sequential descriptions for video frames. You excel in identifying and conveying changes in actions, behaviors, environment, states and attributes of objects, and camera movements between adjacent video frames.

# Skills
## Skill 1: **Describing Object Actions and Behaviors**
- Describe the action or behavior of objects within the frame.
- Notice and describe changes in the actions or behaviors between frames.

## Skill 2: **Describing Environment and Background Variations**
- Elaborate on environment and background alterations seen between frames.

## Skill 3: **Describing Object Appearances**
- Describe the appearance of objects within the frame.
- Depict variations in the states and attributes of objects between frames.

## Skill 4: **Describing Camera Movements**
- Perceive the camera's movements, such as panning or zooming.
- Convey these camera movements and describe how they impact the footage displayed.

# Constraints
- State facts objectively without using any rhetorical devices such as metaphors or personification.
- Stick to a narrative format for descriptions, avoiding list-like itemizations.
- Exclude sounds-related aspects, given the unavailability of audio signals.
- Descriptions should be fluent and precise, avoiding analyzing and waxing lyrical.
- Descriptions need to be concise, describing only the information that can be determined, without analysis or speculation.
- If there is only one inputted frame, that is, the first frame, describe the image details directly and do not concern yourself with connections between other frames.
- Do not mention the frame number and timestamp of the current frame.

# Structured Input
Video frame <idx_n-1> at <timestamp_n-1> Second(s) <frame_n-1><diff_caption_n-1>
Video frame <idx_n> at <timestamp_n> Second(s) <frame_n>

Figure 7: Differential Caption Prompt Template

**The Summary Prompt Template**

# Character
I will provide you with the descriptions of each of multiple consecutive frames in a video, each containing the content of the current frame and how the current frame has changed relative to the previous frame. Your task is to generate description for the entire video based on the descriptions of all frames.

# Skills
- Summarize sequentially, maintaining coherence between frames and the integrity of the timeline.

# Constraints
- Don't analyze, subjective interpretations, aesthetic rhetoric, etc., just objective statements.
- Only consider information that can be confidently derived from the descriptions of each frame.
- Do not extrapolate or imagine, remove uncertain information.
- No mention of specific frames index or timestamps.

# Structured Input
<timestamp_1><caption_1><timestamp_2><caption_2>...<timestamp_n><caption_n>

Figure 8: Summary Caption Prompt Template

## A.4 Comparison of Caption Quality

Table 6: **Comparison of lexical composition of the captions** generated by GPT4V and ShareCaptioner-Video.

| Lexical Category | n. | adj. | adv. | v. | num. | prep. |
|---|---|---|---|---|---|---|
| GPT4V | 27.5% | 11.2% | 2.0% | 12.3% | 0.3% | 11.3% |
| Share-Captioner | 28.1% | 2.8% | 1.5% | 12.2% | 0.5% | 11.4% |

Table 7: **Human preference** on ShareCaptioner-Video vs. GPT4V over 100 validation samples and 10 volunteers.

| | GPT4V | ShareCaptioner-Video | Comparable |
|---|---|---|---|
| Percentage | 37.8% | 36.5% | 23.7% |
| Avg. Score | 2.2 | 2.0 | - |

**Quantitative captioning capability comparison between ShareCaptioner-Video and GPT4V.** We first analyze the linguistic composition of the captions produced by GPT4V and ShareCaptioner-Video, and the results are presented in Table 6. The analysis reveals that the captions generated by our ShareCaptioner-Video contain a comparable information level to those generated by GPT4V. Furthermore, as shown in Table 7, we first generate 100 captions with GPT4V and ShareCaptioner-Video and then invite 10 volunteers to evaluate the captions based on three aspects. These aspects include (1) **Omission and Fabrication** - checking for no key elements missing in the caption and identifying imaged elements not present in the video; (2) **Distortion** - assessing the accuracy of element attributes such as color and size; and (3) **Temporal Mismatch** - evaluating whether the description accurately reflects the evolution of temporal events in the video. Each pair can earn a maximum of 3 points, one for each criterion met. As anticipated, our ShareCaptioner-Video performs on par with the GPT4V.

**Qualitative comparison of caption quality from various sources.** In Figure 9, we compare the quality of captions from different sources. The result which uses the Panda-Student [14] model learned from multiple LVLMs, generates one-sentence captions for videos. While this approach minimizes errors, it provides insufficient textual information for information-dense videos, hindering the alignment of video and language modalities. The Video ChatCaptioner [10], on the other hand, uses ChatGPT to construct questions, employs BLIP2 [34] for VQA, and then uses ChatGPT again to summarize a complete caption based on the QA. Although this method produces longer captions than Panda-Student, the extensive involvement of ChatGPT often leads to over-imagination and hallucinations, negatively impacting the alignment between video and language modalities. Thanks to our carefully designed pipeline for generating high-quality video captions, GPT4V demonstrated impressive captioning capabilities. Additionally, after fine-tuning on the ShareGPT4Video dataset, our ShareCaptioner-Video also exhibits captioning capabilities comparable to GPT-4V.

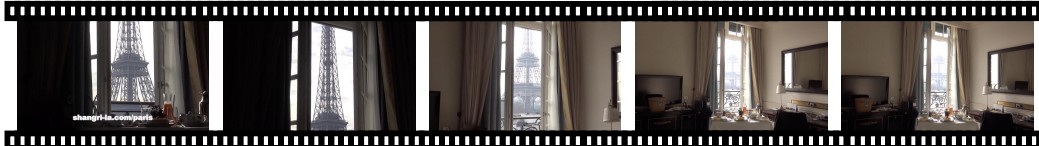

**Panda:** The hotel room has a beautiful view of the eiffel tower and breakfast is set up on the table.

**Video ChatCaptioner:** In the video, a woman is sitting at a table indoors. She is alone at the table and there are no visible objects on the table. The woman appears to be sitting still and there is no visible movement outside the window. There are no other people in the room and no visible decoration or artwork on the walls. The room is well lit and there are no plants or flowers present.

**GPT4V:** The video presents a progressively closer view of the Eiffel Tower from an interior perspective, beginning with a broad shot capturing the tower through a window or glass door, alongside a window sill or table with various items including a glass pitcher or vase and some glasses.

As the video progresses, the focus narrows down on the tower's metallic lattice structure and observation decks, with the interior elements like window frames and curtains framing this iconic Parisian landmark consistently observed.

Then, camera's zoom out reveals more of the room, including a decorative wrought iron or balcony railing, suggesting the viewpoint is from an apartment or room overlooking the tower.
Further pulling back displays more of the room's interior, including a table adorned with glasses, decanters, and decorative pieces, hinting at a dining or living area setting, and a mirror on the wall, adding to the residential ambiance. And then the camera makes a brief pause.

Throughout these transitions, the overcast sky and framing of the Eiffel Tower remain constant, with subtle shifts in sunlight casting changing shadows and light patterns in the room, indicating a progression of time. The positioning of furniture and decorative items remains unchanged, maintaining a consistent interior setting against the backdrop of the iconic Eiffel Tower.

**ShareCaptioner-Video:** The video begins with a scene set in a well-lit room during the daytime, offering a picturesque view of the Eiffel Tower through an open window. The initial setting is cozy, featuring a breakfast nook with a dining table, chairs, and a teapot on a sideboard. The décor is homely is homely, with a dark-colored curtain partially drawn to one side, allowing ample light into the space and highlighting the serene ambiance created by the presence of the iconic landmark outside.

As the video progresses, the camera moves closer to the window, focusing more intently on the Eiffel Tower's intricate lattice structure. With this shift, the details of the room become less discernible, overshadowed by the magnified view of the Tower, which dominates the frame, drawing all attention to its architectural grandeur as seen through the window.

Subsequently, the camera adjusts to include more of the room within the frame, balancing the view of the Tower with the interior details. This broader perspective reveals a neatly arranged room, possibly a dining or sitting area, accented by elegant décor and a selection of items on a sideboard, all bathed in natural light that creates a warm atmosphere. The adjustment in camera positioning allows for a harmonious blend of the indoor setting with the external view, encapsulating the juxtaposition of architectural magnificence against the backdrop of everyday life.

The focus then shifts entirely to the interior, providing a detailed view of the room's layout and design. The Eiffel Tower, while still visible through the window, becomes a secondary element to the domestic setting...

Finally, the scene remains consistent with the previous interior view, showing no significant changes in the arrangement or lighting of the room...

Figure 9: A qualitative comparison of caption quality from various sources. Mistakes within the captions are highlighted in **red**, whereas detailed and accurate parts are emphasized in **blue**.

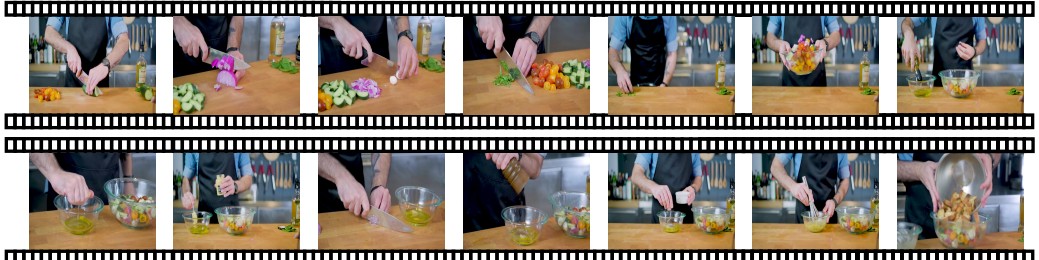

**Summarized Caption:** The video documents a meticulous process of salad preparation by an individual with tattooed forearms, in a well-equipped kitchen. Initially, he is seen slicing a cucumber on a cutting board, with a small pile of multi-colored cherry tomatoes to the side and a bottle of "Grandma's Choice White Wine Vinegar" nearby. Progressively, he shifts to cutting a red onion, then moves on to chopping garlic and finally slicing a green, leafy herb. The tomatoes, cucumber slices, and chopped onions remain untouched during these tasks, emphasizing an organized approach to preparing the dish.

Following the chopping, the man starts to arrange the ingredients on the cutting board, indicating a transition to combining the salad components. At this stage, he holds a large glass bowl filled with brightly colored tomatoes and prepped ingredients, ready for further action. Soon, he continuously tips the glass bowl to ensure the ingredients are evenly mixed.

The focus then shifts to preparing a salad dressing; he pours olive oil into a smaller bowl and proceeds to add lemon juice, followed by scooping mustard into the mixture, then he shifts back to chopping, this time a shallot or red onion, next to the oil-mustard mixture. The scene broadens to show more of his environment and then zooms in as he starts to mix the dressing with another ingredient. He seasons the mixture with black pepper from a wooden grinder, sprinkles with a powder resembling salt, and then begins to whisk vigorously, aiming for an emulsified consistency.

Finally, he adds croutons from a metal bowl into the large glass bowl containing the salad. Throughout, the kitchen setting remains consistent, with the man's actions focused on crafting a vibrant and texturally diverse salad.                                    Correct 😊

Figure 10: The full video caption generated by our DiffSW, with correct temporal understanding and comprehensive detail description.

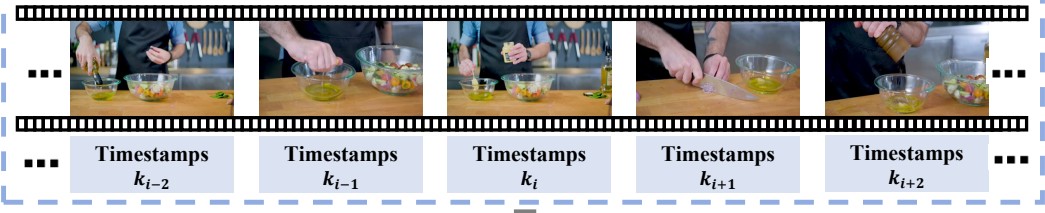

| Timestamps $k_{i-2}$ | Timestamps $k_{i-1}$ | Timestamps $k_i$ | Timestamps $k_{i+1}$ | Timestamps $k_{i+2}$ |

**Multi-frame with Timestamps Caption:** The video maintains a concentrated focus on food preparation from slicing and chopping to blending and dressing, ending with the assembly of a vibrant and texturally diverse salad.In the video, a man with tattooed forearms meticulously prepares a salad in a well-equipped kitchen. He begins by slicing a cucumber, then moves on to cutting a red onion, chopping garlic, and slicing a green, leafy herb. Throughout these tasks, the tomatoes, cucumber slices, and chopped onions remain neatly arranged, highlighting his organized approach.

After completing the chopping, he arranges the ingredients on a cutting board and transitions to combining them in a large glass bowl. He mixes the ingredients together to ensure an even distribution of flavors before preparing the salad dressing. He pours olive oil into a smaller bowl, adds lemon juice, mustard, black pepper and powder resembling salt, and then whisk vigorously for an emulsified consistency.

Returning to chopping, he prepares a shallot or red onion next to the oil-mustard mixture. Finally, he adds croutons from a metal bowl into the salad, signaling the completion of the preparation process.

Throughout the video, the kitchen setting remains consistent as the man focuses on crafting a meticulous and flavorful dish. The video maintains a concentrated focus on the food preparation, from slicing and chopping to blending and dressing, ultimately resulting in the assembly of a vibrant and texturally diverse salad.

Temporal Mismatch 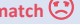

Figure 11: The video caption generated by GPT4V with multiple frames alongside timestamps as input. The GPT4V failed to understand the frames with the correct temporal order and outputs an incorrect caption.

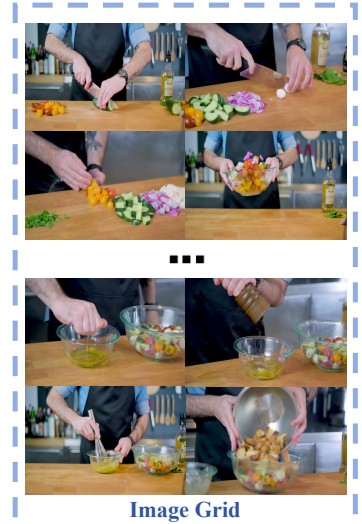

**Image Grid**

**Image Grid Caption:** The video depicts a detailed salad preparation process by a man with tattooed forearms in a well-equipped kitchen. He begins by slicing a cucumber on a cutting board, then chopping garlic. Each ingredient remains separate during preparation, highlighting an organized approach.

Afterward, he arranges the ingredients in a large glass bowl and mixes them together evenly. He then prepares a salad dressing by combining olive oil, lemon juice and whisking vigorously for an emulsified consistency.

Lastly, he adds croutons from a metal bowl to the salad, indicating the final step. Throughout the process, the kitchen setting remains consistent, with the man's actions focused on creating a meticulous and flavorful dish. The video maintains a concentrated focus on food preparation from slicing and chopping to blending and dressing, ending with the assembly of a vibrant and texturally diverse salad.

Description Omission 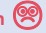

Figure 12: The video caption GPT4V generated by concatenating all the frames into a large image. Some video details are missing.

## A.5 Data Statistics

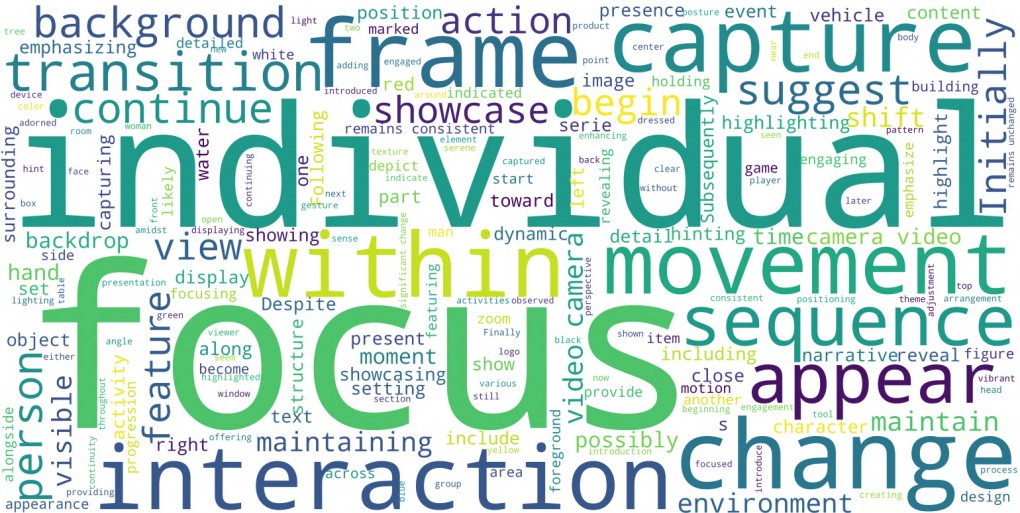

Figure 13: Word cloud of the captions in the Share4Video dataset.

| Data Source | Samples | Total Time(hours) | Avg. Length(#word) |
|---|---|---|---|
| Mixkit | 56k | 42.0 | 104.8 |
| Pixabay | 652k | 353.3 | 102.5 |
| Pexels | 4104k | 2561.9 | 100.5 |
| Total | 4812k | 2957.2 | 102.6 |

Table 8: Statics of 4.8M high-quality video-caption pairs generated by our ShareCaptioner-Video.

## A.6 Pseudo Code

```python
# Input:  video_list, - Video list to be processed
#         caption_feature_list, - Temporary caption features list
#         threshold       - Caption-caption similarity threshold
#
# Output: diverse_video_pool - Selected videos

import torch

# Init result array
diverse_video_pool = []

# Semantically Rich Video Caption Selection
for video in video_list:
    # Obtain the features of caption using BERT
    caption_feature = get_bert_cls_token(video.caption)

    # Calculate the similarity between captions
    sim_matrix = caption_feature * torch.stack(caption_feature_list)

    # Add results to list if maximum similarity is below threshold
    if torch.max(sim_matrix.sum(dim=1)) < threshold:
        caption_feature_list.append(caption_feature)
        # Accommodate videos with diversity
        diverse_video_pool.append(video)
```

Figure 14: Pseudo code of semantic-based data filtering.

```python
# Input:  video,          - Video to be processed
#         time_interval,  - Minimum time interval between keyframes
#         threshold       - Inter-frame similarity threshold
#
# Output: key_frame_pool - Extracted keyframes

# Minimum frame interval
frame_interval = time_interval * video.fps
# All frames of the video
video_frame_list = video.frame_list

# Initialize the keyframe pool using the first frame of the video
key_frame_pool.append(video_frame_list[0])

# Semantic-aware keyframe extract
for idx in range(1, len(video_frame_list), frame_interval):
    frame_feature_1 = get_clip_cls_token(key_frame_pool[-1])
    frame_feature_2 = get_clip_cls_token(video_frame_list[idx])
    similarity = similarity(frame_feature_1, frame_feature_2)
    if similarity < threshold:
        key_frame_pool.append(video_frame_list[idx])

# Accommodate videos with small content changes
if len(key_frame_pool) == 1:
    key_frame_pool.append(video_frame_list[-1])
```

Figure 15: Pseudo code for semantic-aware key-frame extraction.

