# OpenReview forum: "ShareGPT4Video: Improving Video Understanding and Generation with Better Captions"
_NeurIPS.cc/2024/Datasets_and_Benchmarks_Track — NeurIPS 2024 Track Datasets and Benchmarks Poster_

### Official Review · Reviewer_R7XY · 2024-07-23
**ShareGPT4Video: Improving Video Understanding and Generation with Better Captions**

**Rating:** 7
**Confidence:** 4
**Correctness:** Yes, the experimental procedure is so…

**Review:**

It is very good work, offering a new dataset to the community, and a novel differential captioning technique which seems to help with temporal understanding. More analysis needs to be done about this mechanism.

**Strengths:**

The paper offers several key insights to the NeurIPS community. Whereas there are situations when training on generated data leads to degraded performance, specifically image generation, it seems that in the case of NLP, generated content can help with training data, avoiding the time and cost of human annotators. The authors use this to great effect and show through careful prompting that common difficulties in current systems may be overcome. To that effect, they are offering a large, high-quality, balanced dataset for video understanding research, which will be valuable to the community. They also offer a methodology for change detection which helps the community see that this can work and with good results. Additionally, they train a video generation model on the high-quality dataset, showing that improvements in the captions, qualitatively improve the generated videos.

**Additional Feedback:**

Aside from the above comments, this was a very good paper.
One minor comment: you use the word “statics” in two places. Is this a new way of referring to “statistics”?

**Clarity:**

The paper is well-written and easy to follow and understand. However, there are some suggestions that would make to help make it clearer.
One suggestion is to clear up the redundancy in the abstract (lines 18-27 repeat the info in lines 4-9).
Another suggestion (or consideration) is the naming the offerings. It was hard for me, as a reading to make sense of the ShareGPT4Video, ShareCaptioner-Video, and ShareGPT4Video-8B. I had to keep a cliff notes and keep referring back to it to learn them. ShareGPT4Video is a dataset, but ShareGPT4Video-8B is a LVLM. ShareCaptioner-Video is used in two ways: differential captions and holistic captions.
A third suggestion for clarity, would be to move Appendices 7 and 8 into the main manuscript. These sections are about related work and limitations.
Additionally, consider discussing more limitations and ethical impacts. These sections are good but it would seem there are many more implications here.

**Documentation:**

Yes, there are enough details to recreate the results.

**Ethics:**

None beyond what the authors already mention.

**Limitations:**

The main novel contribution in this work was the differential sliding window approach to captioning and then using those differential captions in prompting for a comprehensive caption for a whole video. The results show that this works well, but there is not enough discussion about the trade-offs involved, and other possible solutions (to solving temporal and coherence issues) considered. What other methods were considered? Why does this one work? What could make it better? How is it effective? More analysis around this novel idea and exploration of it’s utility would be helpful to the community.

**Opportunities For Improvement:**

The results show that this works well, but there is not enough discussion about the trade-offs involved, and other possible solutions (to solving temporal and coherence issues) considered. What other methods were considered? Why does this one work? What could make it better? How is it effective? More analysis around this novel idea and exploration of it’s utility would be helpful to the community.

**Relation To Prior Work:**

This is clearly discussed but it in the appendix. Please move to main part of manuscript.

**Summary And Contributions:**

The authors research Video Understanding and Video Generation and make a series of contributions to the field of study. They use GPT4V heavily to aid in the development of their series. They hypothesize that high-quality annotations will aid in both tasks (understanding and generation) and that current systems fail at identifying temporal changes and lead to confused results. They develop a systematic way to make “differential” captions, using a sliding window, between two frames in a video, and a template, and aided by GPT4V. Then, they send all of the differential captions for a video to GPT4V with another template, to obtain a comprehensive caption for the whole video. They say the differential sliding-window approach allows the system to focus on the temporal changes between frames, can be preprocessed ahead of time and reused, and could be suitable for streaming applications. They run several experiments to demonstrate efficacy. The research offers three main contributions to the community: 1. A dataset suitable for studying video understanding, 2. A captioning model, and 3. An LVLM model that surpasses state of the art in current video understanding benchmarks.

---

> ### Author Rebuttal · Authors · 2024-08-17
>
> We thank you for the positive comments on the contribution of this work and the novel differential captioning technique we proposed. We detail your concerns and our corresponding responses below:
>
> **Q1: The results show that this works well, but there is not enough discussion about the trade-offs involved, and other possible solutions (to solving temporal and coherence issues) considered. What other methods were considered?**
>
> **A1:** Thank you for your valuable feedback. In Figures 11 and 12 of the appendix, we present other considered solutions but omit detailed discussions. We will discuss this here and commit to adding these to the supplementary materials in the next version.
> While designing the video captioning pipeline, we explore several approaches, with two described in detail. The first approach involves directly inputting multiple keyframes along with their corresponding timestamps into GPT-4V in an interleaved format. As shown in Figure 11 of the appendix, this approach can accurately describe the details within each frame but often fails to maintain correct temporal consistency. This results in captions that confuse the sequence of events, which is critical for video understanding.
>
> The second approach involves arranging the extracted keyframes into an image grid, following a top-to-bottom, left-to-right sequence, and then inputting this grid along with a prompt into GPT-4V. As depicted in Figure 12 of the appendix, this method improves temporal consistency. However, combining multiple sub-images into a single large image significantly compresses the resolution of each frame, making it difficult for the model to discern fine details, which can limit the effectiveness of video understanding.
> Given these challenges, we sought to explore a solution that ensures both temporal consistency and the ability to capture detailed content from each frame.
>
> **Q2: Why does this one work? What could make it better? How is it effective? More analysis around this novel idea and exploration of it’s utility would be helpful to the community.**
>
> **A2:**  Since the images we input during the sliding process are high-resolution, the details within each frame are captured and described as fully as possible. Additionally, because we focus on the content changes between just two adjacent keyframes during the sliding process, rather than being constrained by the video's length, this straightforward task avoids any temporal confusion. In the captioning pipeline, we control a keyframe window with a fixed size of two, where each time we input the current keyframe along with the previous keyframe, allowing the model to summarize the changes between the two frames. We then use an LLM to summarize all differential captions. Additionally, the differential caption of the previous keyframe is also input to provide complementary textual information and temporal context from two frames prior. This approach ensures that the resolution of each frame remains high and avoids temporal confusion regardless of the video length.
>
> One potential improvement in the current strategy is that the sliding window size is fixed. For videos with significant camera movement, a longer window is needed to capture these substantial changes. Therefore, future work could involve developing a method that adjusts the window size based on video content to enhance the captioning process.
>
> **Q3: One suggestion is to clear up the redundancy in the abstract (lines 18-27 repeat the info in lines 4-9).**
>
> **A3:** Thanks for your valuable suggestion. We will remove the paragraph "The series comprises: 1) ShareGPT4Video...SOTA performance on three advancing video benchmarks." in lines 3-9 to reduce the redundancy in the abstract. Your feedback has indeed helped us improve the readability of the abstract!
>
> **Q4: Another suggestion (or consideration) is the naming the offerings. It was hard for me, as a reading to make sense of the ShareGPT4Video, ShareCaptioner-Video, and ShareGPT4Video-8B. I had to keep a cliff notes and keep referring back to it to learn them. ShareGPT4Video is a dataset, but ShareGPT4Video-8B is a LVLM. ShareCaptioner-Video is used in two ways: differential captions and holistic captions.**
>
> **A4:** We apologize for any inconvenience our resource naming may have caused during your review. We will add a "VL" middle name to the multimodal model ShareGPT4Video-8B to avoid confusion. Specifically, "ShareGPT4Video" without any suffix will refer to our dataset, "ShareGPT4Video-VL-8B" will refer to our LVLM, and "ShareCaptioner-Video" will refer to our captioner.  Thank you for your careful review and valuable suggestions for improvement.
>
> **Q5: A third suggestion for clarity, would be to move Appendices 7 and 8 into the main manuscript. These sections are about related work and limitations. Additionally, consider discussing more limitations and ethical impacts. These sections are good but it would seem there are many more implications here.**
>
> **A5:** Thank you for your valuable feedback. In the next version, we will move Sections A.7 and A.8 from the appendix to the main text. Additionally, we have included more extensive discussions on limitations and ethical impacts, as shown below:
>
> **Limitations (new added):**  Additionally, the sampling interval for initial sparsification of the original videos and the window length setting in DiffSW was empirically set to 2 seconds and adjacent 2 frames based on the majority of videos. We plan to make these hyperparameters adaptive to video content in future work to handle a wider variety of video content effectively.
>
> **Social impacts (new added):** Our models can be manipulated or "jailbroken" to produce outputs that are non-inclusive or disrespectful as many LVLMs do. This vulnerability highlights the importance of continuing to improve the robustness and ethical alignment of LVLMs to prevent misuse and ensure they contribute positively to diverse applications.

---

### Official Review · Reviewer_74Ty · 2024-07-24
**Response to the authors**

**Rating:** 7
**Confidence:** 5
**Correctness:** Yes
**Clarity:** Yes

**Review:**

1. The paper identifies a relevant and timely challenge of requiring high-quality video-caption data for enhanced understanding and generation.

2. The proposed pipeline is thorough and reasonable. I like that the paper proposes various strategies for semantic-based data filtering, key-frame extraction, and differential sliding window captioning.

3. The captioning design is very comprehensive – fast captioning, sliding captioning, clip summarizing, and prompt re-captioning.

4. The experimental usefulness of the proposed data is quite good – as a replacement to existing datasets, and to train capable video understanding models.

**Strengths:**

Mentioned in the review

**Additional Feedback:**

No

**Documentation:**

Yes

**Limitations:**

Yes

**Opportunities For Improvement:**

1. How much money did the authors spend on the GPT4 based data extraction for 40K instances?

2. What is the drop in the data quality (40K GPT-4V based data versus ShareCaptionerVideo 4.8M)? The paper will benefit from some human evaluations on the quality of the captions.

3. There are no evaluations to understand the quality of different capabilities of the sharecaptioner model (fast captioning, …, prompt re-captioning).

4. It is unclear why the authors use Sliding captioning and clip summarizing mode for generating captions for 4.8M videos. Why did the authors not use fast captioning for fast caption generation?

5. Is there a mode of ShareGPTvideo that can input all the keyframes as an input (but does not concatenate all keyframes into a single image) and output the video caption directly? To train much mode, you just need the input video and the final GPT-4 written caption. This mode should be the most straightforward method. This gives rise to the question whether IXC2-4KD handle videos natively or is it a multi-image language model?

6. The experiments for the text-to-video generation do not have any real evaluations (https://arxiv.org/abs/2311.17982, https://arxiv.org/pdf/2406.03520, https://arxiv.org/pdf/2406.15252). The paper focuses on T2V generation for complex prompts, however, but it is not clear if the finetuned model can understand simple prompts or not.

**Relation To Prior Work:**

Yes

**Summary And Contributions:**

The paper argues that the high-quality captions data is essential for capable video understanding and generation. To solve this, the paper proposes an effective video captioning strategy to focus on inter-frame temporal change understanding, intra-frame detailed content description, and length-agnostic scalability. Specifically, the methodology (DiffSW) captions the first key frame of the video, then captions the difference between the two consecutive keyframes. Based on this method, the paper constructs 40K high-quality data using GPT-4V. Further, the authors train a video captioning model, ShareCaptioner-Video, for generating video captions at scale (4.8M examples). Finally, the authors show that this video-caption data leads to better video understanding and generation models.

---

> ### Author Rebuttal · Authors · 2024-08-17
>
> Thanks a lot for your thorough review and the appreciation of our work. Below, we address your concerns point by point:
>
> **Q1: How much money did the authors spend on the GPT4 based data extraction for 40K instances?**
>
> **A1:** Thank you for your interest in the overall computational cost. Below, we have detailed the cost estimation process. For each sliding window operation, two images with the longest side re-scaled to 1024 pixels and the corresponding prompt are input, resulting in the corresponding output. We estimate that each sliding window's image consumes approximately 1530 tokens (based on a 1024x1024 resolution), the input prompt averages about 350 tokens, and the output token length averages around 160 tokens. Across 40K instances, there were a total of 301,584 sliding and summary captions. With GPT-4 Turbo's input priced at $10 per 1M tokens and output at $30 per 1M tokens, the cost estimation formula is as follows:
>
> $\text{Cost} = 301,584 \times \left(\frac{1530 + 350}{1,000,000} \times 10 + \frac{160}{1,000,000} \times 30\right) \approx \$7,117$
>
> **Q2: What is the drop in the data quality (40K GPT-4V based data versus ShareCaptionerVideo 4.8M)? The paper will benefit from some human evaluations on the quality of the captions.**
>
> **A2:** In Section A.4 of the appendix, we provide a detailed comparison of the captioning quality between our captioner and GPT-4V.
>
> We first analyze the linguistic composition of the captions produced by GPT-4V and ShareCaptioner-Video, with the results presented in Table 6. The analysis shows that the captions generated by ShareCaptioner-Video contain information comparable to that of GPT-4V.
>
> Furthermore, as shown in Table 7, we generated 100 captions using both GPT-4V and ShareCaptioner-Video, and then had 10 volunteers evaluate them based on three criteria. As expected, ShareCaptioner-Video performs on par with GPT-4V.
>
> Qualitatively,  we compare the caption quality from different sources in Figure 9. After fine-tuning on the ShareGPT4Video dataset, ShareCaptioner-Video demonstrates captioning capabilities comparable to GPT-4V. For more details, you can refer to the Section A.4 in the appendix.
>
> **Q3: There are no evaluations to understand the quality of different capabilities of the sharecaptioner model (fast captioning, …, prompt re-captioning).**
>
> **A3:** Thank you for your valuable suggestions. Since there is no tailored metric for evaluating such detailed captions in the video understanding area, we add a data ablation study to help understand the differences in caption quality between the fast captioning and sliding captioning modes. We replace 28K high-quality video-caption pairs from the 181K training data of the ShareGPT4Video-8B model with captions from different sources for the same videos, focusing solely on analyzing the quality of captions.
>
> | Caption Source | TempCompass | VideoBench |  MVBench |   Avg.   |
> |---|:--:|:--:|:--:|:--------:|
> | Panda-Student (Short Caption)|     56.9    |    37.5    |   47.9   |   47.4   |
> | ShareCaptioner-Video (Fast captioning)    |     59.9    |    40.1    |   50.1   |   50.0   |
> | ShareCaptioner-Video (Sliding captioning) |     60.5    |    40.8    |   50.6   |   50.6   |
> | GPT4V |   **61.5**  |  **41.2**  | **51.2** | **51.3** |
>
> As shown in the table, our captioner achieves performance comparable to GPT-4V with significant captioning costs reducing, and outperforming the short captioner with a noteworthy margin.
>
> Within its two captioning strategies, we find the caption quality of the sliding captioning mode is slightly better than the fast sliding captioning mode. Therefore, users could choose the mode based on their specific needs and priorities.
>
> **Q4: It is unclear why the authors use Sliding captioning and clip summarizing mode for generating captions for 4.8M videos. Why did the authors not use fast captioning for fast caption generation?**
>
> **A4:** As we show in A3, the sliding mode has slightly better caption quality, and the computing cost of captioning all the data is acceptable, so we use sliding mode to train the T2V model with better captions and share them with the community.
>
> **Q5: Is there a mode of ShareGPTvideo...or is it a multi-image language model?**
>
> **A5:** Directly extracting multiple frames from a video and feeding them as interleaved images into LVLMs is indeed a more straightforward and intuitive approach. However, before our submission deadline, the leading open-source LVLMs (e.g., LLaVA-Next, InternVL-1.5, IXC2-4KHD) had limited capabilities in multi-image understanding while having superb high-resolution image understanding capability, which benefited from their large-image training data. Given these constraints, we opted to concatenate all the frames into a large image to better leverage the large-image understanding capabilities of the high-resolution LVLMs available at that time.
>
> **Q6: The experiments for the text-to-video generation...but it is not clear if the finetuned model can understand simple prompts or not.**
>
> **A6:** Thank you for your valuable suggestions. Due to time constraints, we only provide quantitative comparison results on VBench. In the table below, we report the average scores across 16 evaluation dimensions, showing that our T2VMs, with the support of high-quality video-caption data, outperform existing competitors. VBench is a comprehensive benchmark, and most of the prompts it uses are short. As a result, our fine-tuned model is also well-equipped to understand and respond to simple prompts effectively.
>
> | Model        | Average Score |
> |--------------|---------------|
> | CogVideo     | 67.0          |
> | VideoCrafter | 73.0          |
> | ModelScope   | 75.8          |
> | LaVie        | 77.1          |
> | Ours         | **77.9**      |

---

> > ### Comment · Reviewer_74Ty · 2024-08-18
> > **Response to Authors**
> >
> > Hi,
> >
> > I thank the authors for answering the questions well.
> >
> > - re: the leading open-source LVLMs (e.g., LLaVA-Next, InternVL-1.5, IXC2-4KHD) had limited capabilities in multi-image understanding
> >
> > While I don't expect the authors to perform this experiment for me, I somewhat disagree with this statement. There are plenty of open-source Video LLM models such as mplugowl-video, video-chatgpt, video-llava etc. they provide a more natural way of working with videos in the long run when the number of frames is too large. The existing trick of creating a bigger image works as far as the number of frames is limited.
> >
> > - no real eval for complex prompts.
> >
> > since the paper is positioned as a dataset that can improve complex prompt understanding, it is imperative that there are real evals to show quantitative numbers. it becomes the burden of the authors to make some reasonable eval to showcase that there method works in the setting it is supposed to. for the revised version, it might be better to curate some complex prompts and show quantitative improvements.
> >
> > Overall, i like the paper and will keep my scores.

---

> > > ### Author Rebuttal · Authors · 2024-09-01
> > >
> > > Thanks again for your valuable feedback!
> > >
> > > - The choice of the base model for the video captioner.
> > >
> > > Video-ChatGPT and Video-LLaVA are still limited by the number of frames they can process. After our submission deadline, several outstanding multimodal models capable of natively supporting long-context inputs have emerged, such as mPLUG-Owl3, InternVL-2, and Qwen2-VL. We plan to select an appropriate model from these to update our ShareCaptioner-Video, further supporting the community. Thank you once again for your valuable suggestions.
> > >
> > > - No real eval for complex prompts.
> > >
> > > We would like to gently remind you that VBench already provides several dimensions for evaluating generative capabilities under complex prompts. We will be including the relevant results in the supplementary materials.
> > >
> > > Finally, thank you once again for appreciating our work and for the valuable feedback that has helped enhance our efforts.

---

### Official Review · Reviewer_6BYE · 2024-07-26

**Rating:** 6
**Confidence:** 5
**Correctness:** Yes
**Clarity:** Yes

**Review:**

This paper introduces a comprehensive and well-constructed video understanding dataset, which makes a timely contribution for the field of LVLMs and T2VMs.

Please see Strengths and Opportunities For Improvement for pros and cons of this work.

**Strengths:**

- The figures are informative and the tables are well-organized.
- The proposed differential video captioning strategy makes a good attempt to address the challenges of inter-frame temporal change understanding and intra-frame detailed content description.
- The ShareGPT4Video dataset is large and diverse, covering a wide range of categories and sources. The captions are detailed and informative.
- The ShareCaptioner-Video model is efficient and effective and supports various features like fast captioning, clip summarizing, and prompt re-captioning.
The ShareGPT4Video-8B model achieves reasonably good performance on three video benchmarks.

**Additional Feedback:**

N/A

**Documentation:**

Yes

**Ethics:**

No ethical concerns

**Limitations:**

Yes

**Opportunities For Improvement:**

- Missing some representative baselines in the comparison, such as PLLaVA [1] and VILA-1.5 [2]. Should include them in the experiments.
- The differential video captioning strategy may not be suitable for all types of videos, such as videos with fast-paced action or videos with complex camera movements. Could you discuss more about the limitations of this approach (e.g. failure cases) and how to address it?
- What is the reason behind using Panda-Student to generate the captions and the CLS token from the Bert-Base-Uncased model to conduct the data filtering? Is the capability of Panda-Student and Bert-Based sufficiently generalizable to determine the similarity? Please explain the rationale of this choice rather than some other models.
- How reliable is the Semantic-aware Key-frame Extraction with CLIP features? Please provide some examples. Also, how is this sliding-key-frame approach compared with clustering on all frames in V and selecting the frame closest to the centroid of each cluster?

[1] PLLaVA : Parameter-free LLaVA Extension from Images to Videos for Video Dense Captioning. https://github.com/magic-research/PLLaVA
[2] VILA: On Pre-training for Visual Language Models. CVPR 2024. https://github.com/NVlabs/VILA

**Relation To Prior Work:**

Yes

**Summary And Contributions:**

This paper introduces ShareGPT4Video, a dataset for video understanding with large video-language models (LVLMs) and video generation with text-to-video models (T2VMs). The collection comprises: 1) ShareGPT4Video, a 40K video dataset with dense captions annotated by GPT-4V; 2) ShareCaptioner-Video, a model for captioning arbitrary videos, along with 4.8M videos annotated with it; and 3) ShareGPT4Video-8B, an LVLM that achieves strong performance on three video benchmarks. The authors introduce a differential video captioning strategy to generate high-quality video captions that can capture temporal information in frames.

---

> ### Author Rebuttal · Authors · 2024-08-17
>
> We appreciate your recognition of our dataset as a comprehensive, well-constructed, and timely contribution to both the LVLMs and T2VMs communities. We have endeavored to address your concerns as follows:
>
> **Q1:Missing some representative baselines in the comparison, such as PLLaVA [1] and VILA-1.5 [2]. Should include them in the experiments.**
>
> **A1:** Both of these recent (April and May 2024) and solid works that contribute to the field of video understanding, and we will include discussions on them in the related work section of the next version. The primary goal of our work is not to achieve absolute top performance but to explore and highlight the importance of high-quality video-caption data in the fields of video understanding and video generation. We also aim to introduce a series of high-quality datasets and captioners to advance the community.
>
> We provide a detailed comparison of three video understanding benchmarks in the table below. As seen in the table, our model, supported by high-quality video-caption data, achieves excellent performance with minimal data. We obtained the best results in two out of three benchmarks.
>
> | Model             | Training Data | TempCompass | VideoBench |  MVBench |   Avg.   |
> |-------------------|---------------|:-----------:|:----------:|:--------:|:--------:|
> | PLLaVA-7B         | ~2M           |     53.3    |    41.4    |   46.5   |   47.1   |
> | VILA-1.5-8B       | **~51M**      |     59.2    |  **54.3**  |   46.2   | **53.2** |
> | ShareGPT4Video-8B | ~1.5M         |   **61.5**  |    41.2    | **51.2** |   51.3   |
>
> **Q2: The differential video captioning strategy may not be suitable for all types of videos, such as videos with fast-paced action or videos with complex camera movements. Could you discuss more about the limitations of this approach (e.g. failure cases) and how to address it?**
>
> **A2:** Good point! In constructing our annotation process, based on the characteristics of most videos, we sample the video frames with a 2-second interval and remove the redundant frames with the proposed semantic-aware keyframe extraction strategy. For fast-paced videos, we can reduce the sampling interval (e.g., to 0.5 seconds) for better understanding.
>
> When there is complex camera movement, the sliding window with only two adjacent keyframes may not provide sufficient context for the model to understand the video correctly. In such cases, increasing the window length can help mitigate the issue. However, determining the appropriate window length during the sliding process is challenging and involves a trade-off between effectiveness and computational cost. Although we chose the window length as 2 in this work for most videos, developing an adaptive method to determine the window length could better handle more challenging videos in future work.
>
> **Q3: What is the reason behind using Panda-Student to generate the captions and the CLS token from the Bert-Base-Uncased model to conduct the data filtering? Is the capability of Panda-Student and Bert-Base sufficiently generalizable to determine the similarity? Please explain the rationale of this choice rather than some other models.**
>
> **A3:** The goal of data filtering is to select videos of diverse types and topics videos from large-scale candidates. It requires basic video understanding and semantic extraction capability, and we find the two models are sufficient to handle it effectively. In detail:
>
>  - For video understanding, the Panda-Student is trained on the most relevant caption data from 70 million videos using multiple multimodal teacher models, giving it strong generalization capabilities to generate brief captions for diverse videos.
>  - For semantic extraction, the BERT-Base is pre-trained on a large corpus of English data in a self-supervised fashion. The model learns an inner representation of the English language that is widely used to extract features useful for downstream tasks. We find the similarity score of its CLS token is representative of distinguishing video captions about different types and topics efficiently.
>
> **Q4: How reliable is the Semantic-aware Key-frame Extraction with CLIP features? Please provide some examples. Also, how is this sliding-key-frame approach compared with clustering on all frames in V and selecting the frame closest to the centroid of each cluster?**
>
> **A4:** Good point!
> 1. The large-scale image-text contrastive training enables the CLIP vision encoder to focus on the semantics of the images and neglect the unnecessary detail differences between similar images, providing a reliable measure of inter-frame similarity.
>
>     **In the attached pdf file**, we compare Semantic-aware Key-frame Extraction with the traditional optical flow method [1] with a representative case, demonstrating that our approach is more effective and robust.
>
>     The video has slow camera movement and lighting fluctuation, while its content is almost unchanged for humans. The Optical flow is sensitive to the changes and generates redundant key-frames, while our method is robust toward it.
>
> 2. The clustering and selecting method does not account for temporal information. If similar scenes appear multiple times in different parts of a video, this keyframe extraction method can disrupt the chronological order, leading to incorrect temporal understanding.

---

### Official Review · Reviewer_e5xH · 2024-08-03

**Rating:** 6
**Confidence:** 3
**Correctness:** Yes
**Clarity:** Yes

**Review:**

Pros:
1. This work focuses on improving video understanding and generation by providing better captions. The problem to be studied is explained very clearly.
2. The related works are sufficiently discussed and well-organized.
3. This paper is written with a clear structure and has intuitive figures to illustrate the work.

Cons:
1. Why did authors initially filter out videos from our selected data sources longer than two minutes? This operation may introduce bias and harm the generalization.
2. Authors may need to discuss the difference between share-captioner models and proposed sliding captioning. Why proposed method is better?

**Strengths:**

See Pros.

**Additional Feedback:**

N/A

**Documentation:**

Yes

**Limitations:**

Yes

**Opportunities For Improvement:**

See Cons.

**Relation To Prior Work:**

Yes

**Summary And Contributions:**

The ShareGPT4Video series is introduced to enhance the video understanding capabilities of large video-language models (LVLMs) and to improve text-to-video model (T2VM) video generation through the use of dense, precise captions. The series includes:
1) ShareGPT4Video: This component features 40,000 densely captioned videos from various sources, created with a carefully crafted data filtering and annotation strategy.
2) ShareCaptioner-Video: An efficient captioning model that has been used to annotate 4.8 million high-quality aesthetic videos, capable of handling arbitrary videos.
3)ShareGPT4Video-8B: A simple yet highly effective LVLM that has achieved state-of-the-art performance on three advanced video benchmarks.
The development of these tools involved moving away from traditional, costly human annotation methods towards using GPT4V for automated video captioning.

---

> ### Author Rebuttal · Authors · 2024-08-17
>
> We thank you for the positive comments on the contribution and meaningful impact of this work. We detail your concerns and our corresponding responses below:
>
> **Q1: Why did authors initially filter out videos from our selected data sources longer than two minutes? This operation may introduce bias and harm the generalization.**
>
> **A1:** Good point!
> 1. With the proposed differential sliding-window captioning strategy, our method could caption videos with arbitrary length, but not limited to only 2 minutes.
>
> 2. When building the ShareGPT4Video dataset, we opt to filter out video longer than 2 minutes primarily due to the capability limitations of current LVLM and T2VM:
>
>    - For LVLMs, current models have limited capabilities for understanding long videos, with most models only able to process 16-64 frames, and still struggling with understanding the detailed content of minute-level video.
>    - For T2VMs, even the leading model, Sora, could only generate videos of up to 2 minutes. The other open-source T2VM still faces challenges in generating fine details accurately for minute-level video.
>
>    In a word, detailed and accurate video understanding and generation at the two-minute level is still challenging and unsolved for mainstream LVLMs and T2VMs, so we present detailed captions of two-minute videos to facilitate the related study, and leave the longer video captioning in future work.
>
>
> **Q2: Authors may need to discuss the difference between share-captioner models and proposed sliding captioning. Why proposed method is better?**
>
> **A2:** Sorry for the confusion. ShareCaptioner-Video is a **model** to generate captions for videos, while the differential sliding-window captioning is a **strategy** it used to generate captions for videos.
>
>  - The ShareCaptioner-Video is designed as an open-source LVLM alternative of GPT-4V, aiming to obtain large-scale, high-quality video-caption data with minimal cost and satisfactory quality. Our captioner is capable of fulfilling all the roles that GPT-4V performs during the annotation process, including supporting the sliding captioning mode.
>  - The differential sliding-window captioning is our carefully designed video caption strategy, we applied it to GPT4V to build our ShareGPT4Video dataset, and also supported it in our ShareCaptioner-Video.
>
> Therefore, there's no direct comparison relationship between sliding captioning and our captioner. Additionally, our captioner offers three other working modes, allowing users to switch modes based on their specific needs.

---

### Author Rebuttal · Authors · 2024-08-17

We sincerely appreciate all reviewers for your time and efforts in the review. All detailed questions of each reviewer are answered accordingly in each column below. We hope these responses can address the reviewers' concerns adequately. Additionally, we have submitted a PDF attachment comparing the effectiveness of our proposed semantic-aware key-frame extraction with the traditional optical flow-based key-frame extraction method [1].

[1] Two-Frame Motion Estimation Based on Polynomial Expansion. Gunnar Farnebäck, SCIA2003

---

### Decision · Program_Chairs · 2024-09-26

**Decision:**

Accept (Poster)

**Comment:**

I have read the review comments as well as the authors' rebuttal. ShareGPT4Video features 40,000 densely captioned videos from various sources, created with a carefully crafted data filtering and annotation strategy. The proposed pipeline is thorough and reasonable. The constructed dataset is helpful for the research community.
As such, I am recommending acceptance of the paper.